# The Chromosome-Scale Genome of *Chitala ornata* Illuminates the Evolution of Early Teleosts

**DOI:** 10.3390/biology13070478

**Published:** 2024-06-27

**Authors:** Zengbao Yuan, Yue Song, Suyu Zhang, Yadong Chen, Mengyang Xu, Guangyi Fan, Xin Liu

**Affiliations:** 1College of Life Sciences, University of Chinese Academy of Sciences, Beijing 100049, China; yuanzengbao@genomics.cn (Z.Y.);; 2BGI-Qingdao, BGI-Shenzhen, Qingdao 266555, China; songyue@genomics.cn (Y.S.); chenyadong@genomics.cn (Y.C.);; 3BGI-Shenzhen, Shenzhen 518083, China

**Keywords:** teleost, pectoral fins, conserved elements, Osteoglossiformes

## Abstract

**Simple Summary:**

As the most diverse vertebrate group, the unique adaptive expansion of euryhaline fishes is critical to understanding vertebrate evolution. In particular, the high degree of consistency of unique paired appendage structures across the extremely morphologically diverse group of teleost fishes has become a fascinating scientific question. Early teleost fishes provide a critical window into the study of this large taxon. Therefore, this study constructs high-quality chromosome-level genomes of Osteoglossiformes (*Chitala ornata*). It also explores the genomic features of early teleost fishes and traces the unique genetic basis of pectoral fin evolution in teleost fishes at the molecular level, which provides an important basis for understanding the evolution of the origin of early teleosts.

**Abstract:**

Teleosts are the most prolific vertebrates, occupying the vast majority of aquatic environments, and their pectoral fins have undergone remarkable physiological transformations throughout their evolution. Studying early teleost fishes, such as those belonging to the Osteoglossiformes order, could offer crucial insights into the adaptive evolution of pectoral fins within this group. In this study, we have assembled a chromosomal-level genome for the Clown featherback (*Chitala ornata*), achieving the highest quality genome assembly for Osteoglossiformes to date, with a contig N50 of 32.78 Mb and a scaffold N50 of 40.73 Mb. By combining phylogenetic analysis, we determined that the Clown featherback diverged approximately 202 to 203 million years ago (Ma), aligning with continental separation events. Our analysis revealed the intriguing discovery that a unique deletion of regulatory elements is adjacent to the *Gli3* gene, specifically in teleosts. This deletion might be tied to the specialized adaptation of their pectoral fins. Furthermore, our findings indicate that specific contractions and expansions of transposable elements (TEs) in teleosts, including the Clown featherback, could be connected to their adaptive evolution. In essence, this study not only provides a high-quality genomic resource for Osteoglossiformes but also sheds light on the evolutionary trajectory of early teleosts.

## 1. Introduction

Teleost fishes are currently the most diverse group of vertebrates on Earth, with a total of more than 30,000 species. Osteoglossiformes, an early teleost group encompassing Arapaimidae, Gymnarchidae, Osteoglossidae, Pantodontidae, Notopteridae, and Mormyridae, is of particular interest. This group dates to the Jurassic Period, making it a valuable link to the past. Recent research indicates that Elopomorpha and Osteoglossomorpha form a sister clade to all other teleosts [1]. Consequently, studying the genetic basis of physiological traits in Osteoglossiformes is important for exploring the macroevolutionary history of teleosts.

The pectoral fin, a crucial locomotive organ in fishes, traces its origins back to the chondrichthyans [2]. However, following the divergence of Sarcopterygii and Actinopterygii, Sarcopterygii’s plesiomorphy [3,4]. In contrast, the early ray-finned fishes (Actinopterygii) retained numerous skeletal features, including the humerus [5,6]. With the emergence of teleost fishes, the pectoral fin underwent further specialization, abandoning its original accessory structures and evolving into a distinct pectoral fin form [7].

The Clown featherback, a species belonging to the Notopteridae family [8], is predominantly found in Asia, particularly Southeast Asia. Due to the lack of a high-quality genome for the Clown featherback, studying the genetic evolution of this species has been challenging. In this study, we de novo assembled the first chromosome-level genome of the Clown featherback. By combining this genome with published genomic data from other teleosts, we aim to comprehensively detail the genomic features of the early teleosts. Furthermore, we seek to unravel clues pertaining to the evolution of teleost pectoral fins.

## 2. Materials and Methods

### 2.1. Library Construction and Sequencing

The adult specimens of *C. ornata* were bought from an ornamental fish market in Qingdao, China, and their muscle tissues were immediately stored in liquid nitrogen. Then, the genomic DNA was extracted from the muscles of the fishes using NucleoBond HMW DNA KIT (MACHEREYNAGEL, Dueren, Germany). The Agilent 4200 Bioanalyzer (Agilent Technologies, Palo Alto, CA, USA) was used to determine the integrity of the DNA. To construct the PacBio high-fidelity circular consensus sequencing (HiFi-CCS) library, eight micrograms of genomic DNA were sheared and concentrated with AMPure PB magnetic beads (Pacific Biosciences, Menlo Park, CA, USA). Using Pacific Biosciences SMRTbell Template Prep Kit v1.0, each SMRT bell library was constructed. The constructed library was selected by Sage ELF for molecules 11–15 kb in size, followed by primer annealing and binding of SMRT bell templates to polymerases with the DNA Polymerase Binding Kit (Pacific Biosciences, Menlo Park, CA, USA). For HiFi-CCS data assembly, we assembled them using hifiasm version 0.9 and converted them to fasta genome files using gfatools. Sequencing was performed on the Pacific Bioscience Sequel II for 30 h by Annoroad Gene Technology, Beijing, China. We sequenced 43.44 Gb long reads (~50× coverage, N50 read length 18.13 Kb) using HiFi. For Hi-C library sequencing, approximately 1 g living muscle tissue was utilized for DNA extraction and library contraction, according to Wang’s method [9]. Sequencing was performed on a BGISEQ-500 sequencer (BGI, Shenzhen, China), generating 102 Gb of clean Hi-C data.

### 2.2. Genome Assessment

For HiFi-CCS data, we assembled them using hifiasm [10] software and converted them to fasta genome files using gfatools. The final genome size was 837,258,286 bp, and Contig N50 was 32.78 Mb. To generate a chromosomal-level genome assembly of *C. ornata,* high-quality Hi-C data were used for further assembly. Firstly, we used HiC-Pro software (version 2.8.0_devel) [11] with default parameters to obtain valid sequencing data. Then, Juicer (version 1.5), an open-source tool for analyzing Hi-C datasets [12], and the 3D de novo assembly pipeline were used to connect the contigs to chromosomes. To evaluate the genome assembly of *C. ornata*, we aligned sequencing data filtered previously using SOAPaligner (version 2.2) [13]. We also calculated its GC depth to rule out possible biases during sequencing or possible contaminations. Then, the genome completeness was estimated with Benchmarking Universal Single-copy Orthologs (BUSCO, version 3.0.1).

### 2.3. Genome Annotation

For the newly assembled genome, genome annotation was carried out to seek protein-coding genes in two ways: (a) the ab initio gene prediction and (b) the homology-based annotation. For ab initio gene prediction approaches, Augustus [14] and GlimmerHMM [15] were used with *Danio rerio* as the species of HMM model to predict gene models. For homology-based annotation, six species, including *Scleropages formosus*, *Baufortia kweichowensis*, *Danio rerio*, *Oryzias latipes*, *Takifugu rubripes*, and *Gasterosteus aculeatus*, were aligned against the genome assembly using BLAT software (version 0.36) [16] and GeneWise software (version 2.4.1) [17]. Protein-coding genes were obtained by combining the different evidences using Glean software (version 1.0) [18]. In the final gene models, the average length of 28,404 genes was 9244.86 bp. The average length of coding sequences, exons, and introns was 1347.22 bp, 209.59 bp, and 1455.04 bp. To better understand the evolutionary dynamics of genes, gene family expansion and contraction analysis was performed using Cafe (v3.1) software [19].

### 2.4. Identification of Repetitive Sequences

To ensure the comparability of repeat element annotation between different species, we used two approaches to identify repeat elements in the genome: homolog-based prediction and de novo prediction. For homology-based approaches, different types of transposable element sequences in Repbase (version 16.02) [20] were aligned against the assembly using RepeatMasker (version 3.3.0) [21] with parameters “-q -nolow -no_is -norna -engine ncbi”. We used Tandem Repeats Finder (v4.07) to find tandem repeats. RepeatModeler [22] and LTR-Finder (version 1.0.6) with parameters “-C -w 2” and danRer7-tRNAs.fa as tRNA database [23] were used to perform de novo prediction of repeat sequences, and the results were combined as the library for RepeatMasker to identify and classify repeat elements.

### 2.5. Construct Whole Genome Alignments (WGAs)

We used the *C. ornata* genome as a reference, and aligned it with the genomes of 11 other species (*Erpetoichthys calabaricus*, *Amia calva*, *Lepisosteus oculatus*, *Scleropages formosus*, *Esox lucius*, *Arapaima gigas*, *Anguilla anguilla*, *Oryzias latipes*, *Danio rerio*, *Megalops atlanticus*, and *Gasterosteus aculeatuss*) using LastZ version 1.1 [24] with the following parameters: H = 2000, Y = 9400, L = 3000, and K = 3000. All genomes were masked softly before alignment.

### 2.6. Phylogeny Reconstruction

The resulting LastZ result files were combined into a single multiple genome alignment by MultiZ [25], from which we then obtained a total of 1,909,881 bp with no gap sites across all species. We used alignment blocks including all 12 species to construct phylogenetic trees by Raxml [26] using GTRGAMMA mode with 100 bootstrap replicates.

### 2.7. Analysis of Conserved Elements

We intercepted all sequences of the *Gli3* gene in the upstream and downstream 1000 bp range of the human genome (GRCh38.p14), totaling 278,260 bp sequences (Chr7: 41,959,949–42,238,209). We used two different LastZ comparison models. Close alignment was used in lobe-finned fishes (*Latimeria chalumnae*, *Xenopus laevis*, and *Mus musculus*), and distance alignment was used in cartilaginous and ray-finned fishes (*Callorhinchus milii*, *Erpetoichthys calabaricus*, *Polypterus senegalus*, *Acipenser ruthenus*, *Polyodon spathula*, *Atractosteus spatula*, *Lepisosteus oculatus*, *Amia calva*, *C. ornata*, *Megalops atlanticus*, *Danio rerio*, *Esox lucius*, *Oryzias latipes*, and *Takifugu rubripe*). The distant alignment parameters are H = 2000, Y = 3400, L = 6000, and K = 2200, and the close alignments parameters are H = 2000, Y = 9400, L = 3000, and K = 3000 [7]. The alignments of CNEs referred to in the manuscript were manually checked and plotted with VISTA (v1.4.26) [27]. The functional analysis of CNE comes from the website (https://www.encodeproject.org/, accessed on 1 April 2023.).

## 3. Results and Discussion

### 3.1. Genome Assembly and Annotation of a Chromosome-Level C. ornata

To sequence and assemble the *C. ornata* genome, a total of 43.44 Gb (∼50×) PacBio HiFi CCS data were used to assemble 837.26 Mb genomic sequences, containing 126 contigs with a contig N50 of 32.78 Mb and a GC content of 41.92% (Table 1). To anchor the contig sequences to chromosomes, we constructed a Hi-C library and sequenced ∼102 Gb of Hi-C data. About 794 Mb sequences (94.78% of contig-level assembly) were anchored to 21 chromosomes (Figure 1a and Table 1), which was consistent with the previous report on the *C. ornata* karyotype [28]. Finally, by using BUSCO (Benchmarking Universal Single-Copy Orthologs), we found that ∼96.4% of the complete vertebrate BUSCO genes were covered by our assembly (Table 1), providing further evidence for the fine quality of the assembled genome. Compared to previously published fish genomes of Osteoglossiformes, we found that the presently published *C. ornata* genome is the highest quality genome to date (Figure 1b and Appendix A).

We predicted protein-coding genes with combinational annotation methods (de novo prediction and homology-based prediction) in this genome. In the final gene models, the average length was 9244.86 bp, with an average of six exons. The average length of coding sequences, exons, and introns was 1347.22 bp, 209.59 bp, and 1455.04 bp. We identified 129 contracted gene families and 110 expanded gene families (*p* < 0.05) in *C. ornata* (Appendix A). We further enriched the functions of these expanded genes, which were related to immune, redox responses (Appendix A and Appendix A), suggesting that evolutionary changes in these gene families were related to adaptations in the early survival environment of *C. ornata*.

### 3.2. Phylogenetic and Divergence Time Analysis of Early Teleosts

There are many different hypotheses about the phylogenetic relationships of teleost fishes, especially between Osteoglossomorpha, Elopomorpha, and Clupeocephala. The relationship between Osteoglossomorpha, Elopomorpha, and Clupeocephala has been supported by large-scale transcriptomic data [29], but recent studies have shown that Osteoglossomorpha + Elopomorpha acted as a sister group to Clupeocephala to form the Eloposteoglossocephala, which is consistent with the findings of the Asian arowana [30]. Whole-genome comparison data provides invaluable insights and serves as solid evidence for determining accurate phylogenetic relationships among species [31,32,33,34]. In this study, we performed phylogenetic analyses using genome-wide, specific age data. Our results support the latest hypothesis, the Eloposteoglossocephala hypothesis.

We constructed phylogenetic relationships using comprehensive genome-wide comparison data from 12 species, including *C. ornata* (as well as *Erpetoichthys calabaricus*, *Amia calva*, *Lepisosteus oculatus*, *Scleropages formosus*, *Esox lucius*, *Arapaima gigas*, *Anguilla anguilla*, *Oryzias latipes*, *Danio rerio*, *Megalops atlanticus*, and *Gasterosteus aculeatus*). Furthermore, our analysis confirms that Osteoglossomorpha and Elopomorpha constitute sister groups to Clupeocephala (Figure 2a), consistent with previous research [1]. By integrating differentiation time analysis, we determined that teleost fishes originated around the Permian period (~269 million years ago), while the divergence of Osteoglossiformes took place during the Jurassic (~203 million years ago). Previous studies have shown that plate drift affects speciation to some extent. For example, continental drift has led to the speciation of mammals [33,35]. The combination of our data and paleogeological data suggests that plate movement may be associated with the occurrence of teleost fishes (Figure 2b). Therefore, by using genome-wide data analysis, we determined the evolutionary history of Osteoglossiformes, which provides more references for understanding the evolutionary history of early teleost fishes.

### 3.3. Genomic Repetitive Sequences and Conserved Features

We then carried out genome annotation to identify repeats of *C. ornata*. A total of 123,340,324 bp transposable elements (TEs) were predicted, accounting for 16.58% of the genome (Appendix A). DNA transposons were dominated in TEs with a proportion of 8.70% genome assembly, followed by 6.54% long interspersed elements (LINEs) and 4.00% long terminal repeats (LTRs) of the genome (Appendix A). We found that repetitive sequences showed significant contraction in teleost fishes compared to cartilaginous fishes. It has been previously reported that repetitive sequence can effectively mediate genome expansion in lungfish [36,37]. The specific contraction of TEs is consistent with the fact that the genome of teleosts is smaller, and we also found low LINEs content and LTRs content in early teleosts (Figure 3a,b and Appendix A), which we speculate may be the result of the adaptive evolution of teleosts.

Previous studies of vertebrates have shown that the covariance of early ray-finned fishes with modern teleosts is not conserved, suggesting that the teleosts genome underwent a complex and dramatic process of diversification [7,38]. For example, spotted gar are more obviously co-linear with chickens than zebrafish [38]. Based on genome-wide comparison data, we analyzed the conserved evolution of the chromosomes of *C. ornata*. By analyzing conserved fragments of early ray-finned fishes (Reedfish) and modern teleosts (Zebrafish), we found that the *C. ornata* and Reedfish have a good collinearity relationship, which is consistent with their evolutionary status as primitive teleosts (Figure 3c,d), suggesting that the *C. ornata* has more ancestrally conserved genomic features than modern teleost fishes.

### 3.4. Ancient Genetic Regulation Associated with Pectoral Fin Evolution

Various non-coding conserved elements (CNEs) have been reported to contribute to morphological diversification during evolution [7,32,37,39,40]. One example is an CNE downstream of the *Hand2* gene critical for heart development [7]. In addition, CNE, which is located between the exons of genes, also plays an important role. In the study of the genome of African lungfish [37], it was found that the CNE of the *Foxp1* intergene region plays an important regulatory role and is essential for vertebrate lung function. Previous studies have shown that the pectoral fins generate thrust through movement, which affects the swimming behavior of fishes [41]. The pectoral fins of fish are a homologous organ of the forelimbs of tetrapods [42,43,44,45,46,47]. Tetrapods evolved from fins to limbs through the inheritance of limb enhancers from their ancestors and a series of genetic innovations, which laid a crucial foundation for their adaptation to terrestrial environments [7,36,37,48]. However, teleosts lost some of their original pectoral-fin structure, which can otherwise be observed in the fins of non-teleost ray-finned fishes and extant lobe-finned fishes (Figure 4a). Specifically, the pectoral fins of early vertebrates are divided into three parts: propterygium, mesopterygium, and metapterygium [42,49,50]. This structure is still found in cartilaginous fish and some early ray-finned fishes. But lobe-finned fish retained only the metapterygium and developed into the forelimbs of tetrapods, while teleosts completely discarded the metapterygium to form a minimalist fin [42,49]. We hypothesize that other limb skeletal regulatory elements from ancestors may also be present in basal fishes. The *Gli3* gene encodes a protein that belongs to the C2H2-type zinc finger proteins subclass of the *Gli* family. The *Gli3* gene has been reported to be associated with limb development in mice and fin development in fishes [51,52,53] (Figure 4b). Based on our genome-wide comparison dataset, we observed a specific conserved non-coding element located between the eleventh and twelfth exons of the *Gli3* gene, which is present in all gnawed vertebrates except teleost fishes (Figure 4c). To determine CNE function, we validated it in combination with DNase-seq data and found that the CNE overlaps with the open reading frame (ORF) region of *Gli3* (Figure 4d), and our data suggest a regulatory role for this CNE.

Thus, our results suggest that, with the help of a comprehensive comparative analysis of the genome of early teleost fishes, it can be suggested that the loss of metapterygium in teleosts may be related to the differentiation of certain regulatory elements.

## 4. Conclusions

We have assembled a chromosome-level genome of *C. ornata*, a crucial species of early teleosts, with a genome size of 837.26 Mb. This assembly marks the highest quality chromosome-level genome achieved thus far within the Osteoglossiformes order, offering a reference-level genomic resource for the investigation of early teleosts.

Our phylogenetic analysis indicates that teleosts originated approximately 269 million years ago, with *C. ornata* diverging around 203 million years ago. This timing suggests a potential correlation with paleotectonic plate movements, hinting at a possible influence of these geological events on the speciation of Osteoglossiformes.

Furthermore, our study revealed specific expansions and contractions of TEs linked to teleost fishes. These changes in TEs might have contributed to the emergence of a more compact genome structure in teleosts. Additionally, we observed the loss of certain CNEs associated with the *Gli3* gene, specifically in teleosts. Binding experiments further demonstrated that *Gli3* suggests that the loss of metapterygium in teleost fishes may be related to the differentiation of certain regulatory elements. Overall, our comprehensive dataset provides invaluable resources for delving deeper into the evolutionary history of early teleosts.

## Figures and Tables

**Figure 1 biology-13-00478-f001:**
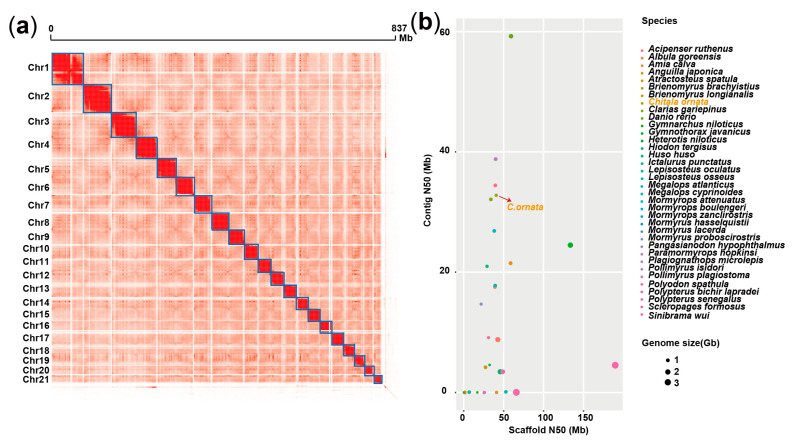
Chromosome-level genome assemblies of *C. ornata*. (**a**) The interaction map of Hi-C data. (**b**) Genomic quality statistics of the Osteoglossiformes.

**Figure 2 biology-13-00478-f002:**
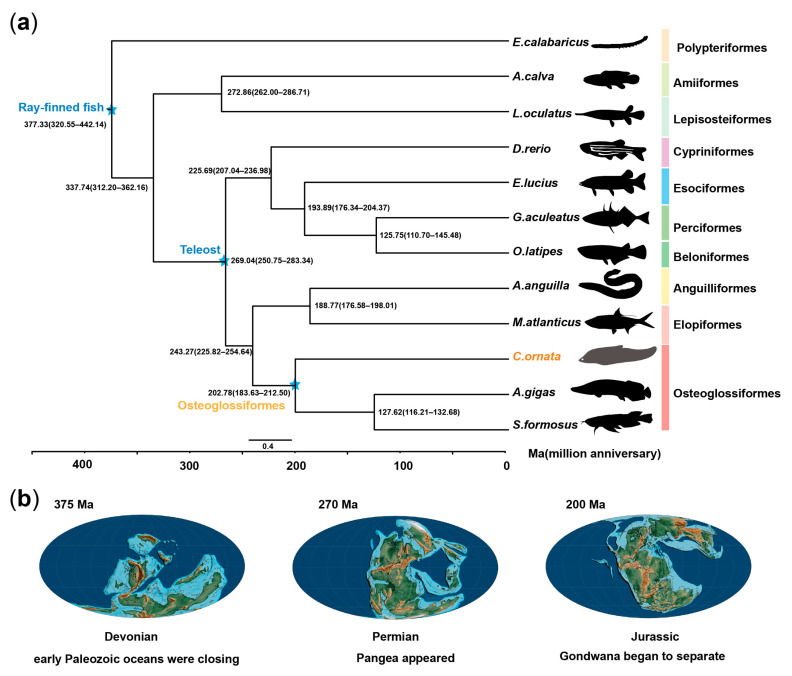
Phylogenetic and divergence time analysis. (**a**) Phylogenetic relationship analysis of *C. ornata*. (**b**) Analysis of paleogeologic changes and divergence times.

**Figure 3 biology-13-00478-f003:**
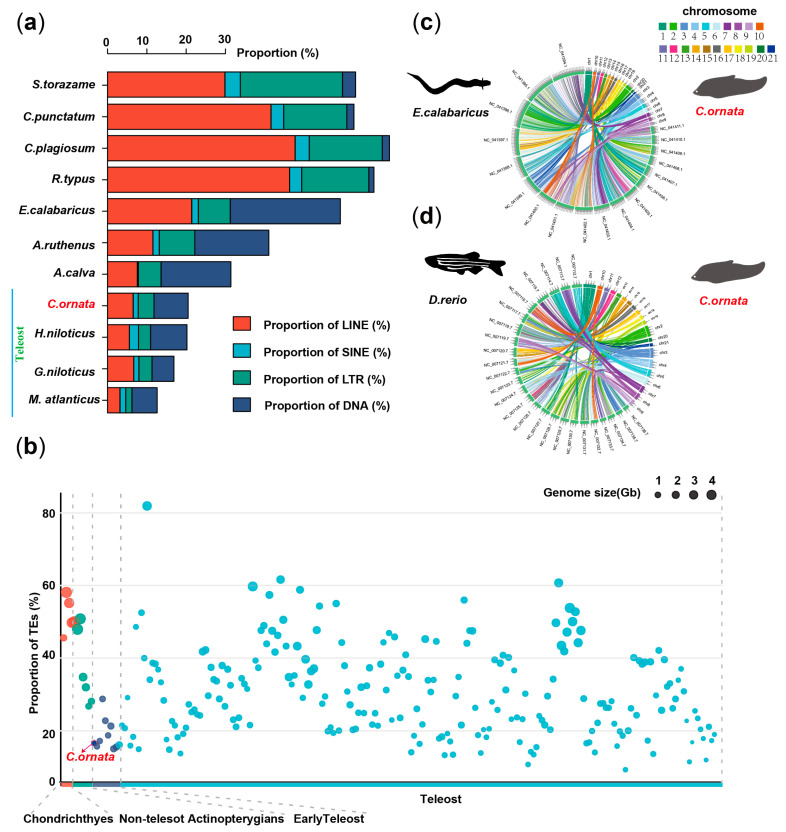
TEs and synteny analysis across multiple species. (**a**) Comparative analysis of TE composition in different fishes. (**b**) Total proportion and genomic characterization of TEs in 235 fishes. (**c**) Genome synteny of the Reedfish–Clown featherback. (**d**) Genome synteny of Zebrafish–Clown featherback.

**Figure 4 biology-13-00478-f004:**
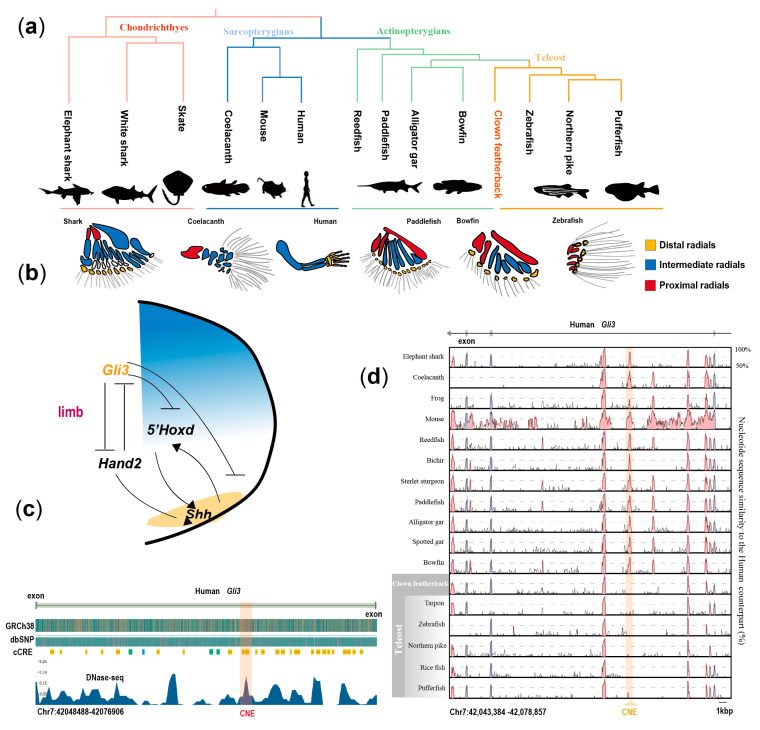
Identification of conserved elements of the pectoral fin. (**a**) Hypothetical transitions in pectoral fin evolution. (**b**) Diagram of *Gli3* contributing to limb development [51]. (**c**) VISTA plot showing the presence of a limb-related CNE intro of the *Gli3* gene across all jawed vertebrates except the teleost. Peaks (blue, exons; red, non-coding regions) indicate regions with conserved sequences compared to their human counterparts. The *Gli3*-CNE is highlighted in pale yellow. (**d**) Human DNase I hypersensitivity site (DHS) data suggest CNE as a potential regulatory element (data from https://www.encodeproject.org/ and accessed on 1 April 2023).

**Table 1 biology-13-00478-t001:** Summary of the species and genome data in this study.

Scientific Name	*Chitala ornata*
English name	Clown featherback
Contig number	126
Contig N50 (bp)	32,781,493
Contig N90 (bp)	10,454,454
Chromosome number	21
Scaffold N50 (bp)	40,727,490
Scaffold N90 (bp)	22,642,485
Hi-C-anchored ratio	94.78%
Assembled genome size (bp)	837,264,786
GC content	41.92%
Genome-complete BUSCOs (C)	96.40%
Complete and single-copy BUSCOs (S)	93.10%
Complete and duplicated BUSCOs (D)	3.30%
Fragmented BUSCOs (F)	1.40%
Missing BUSCOs (M)	2.20%

## Data Availability

The genome assemblies have been deposited in the China National GeneBank DataBase (CNGB) under BioProject CNP0004025.

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
