# Peer review of "The Chromosome-Scale Genome of Chitala ornata Illuminates the Evolution of Early Teleosts"

_biology, 2024, doi:10.3390/biology13070478_

Round 1

Reviewer 1 Report

Comments and Suggestions for Authors

The manuscript "The first chromosome-scale genome of Chitala ornata illuminates the evolution of pectoral fins in teleosts" presents the genome of a species of osteoglossiform and makes inferences about paleobiogeography and the evolution of teleosts. I make some textual corrections directly in the text. I have no objections regarding the applied methodology. However, the authors go too far in exploiting the obtained results.

1.     Phylogenetic relationships: There are hundreds of proposed phylogenies on the evolution of actinopterygii, especially Teleostei. These references are ignored, and only the work of Parey et al. (2023) where Osteoglossomorpha + Elopomorpha are sister groups to Clupeocephala is cited. I remind the authors that only a dozen species were included in the analysis, so the results should be discussed with great caution. There is an enormous number of fossils and morphological synapomorphies pointing to other evolutionary scenarios.

2.     Biogeography: Based on the dating of the obtained topology, the authors claim that the evolution of teleosts has temporal congruence with the separation of continents. Well, since when did the continents stop moving? Continental drift is continuous; at no point in the geomorphological history of the planet has it been particularly greater. It is possible that the drift of continents affected the evolution of teleosts, but how did this happen? How can the authors confirm this relationship without testing any hypotheses? Thus, the results are discussed in overly speculative terms.

3.     Evolution of limbs: I'm not an expert in teleostei development, so I could contribute little to this last section. However, it is striking how superficially the authors address such a relevant topic. The authors state: "we proposed that the absence of this CNE segment in teleost fishes may account for the loss of its primitive skeletal structure, and that conserved regulatory elements are important genetic innovations in the evolution of vertebrate limbs." But what primitive structures were lost? They don't explain anything.

Finally, I emphasize that the publication of the species genome is quite important for understanding the osteoglossiforms’ evolution, but I am skeptical about the presented results as they lack theoretical basis for their propositions. I suggest to the authors that they reorganize the manuscript to highlight the contributions of the work within the reach of the obtained results, avoiding speculative conjectures.

Comments on the Quality of English Language

The quality of English is certainly not the main issue in the paper. The authors express themselves clearly, without major errors. Sometimes, however, there are sentences that were likely translated directly into English and sound somewhat unusual to native readers

Author Response

Comments 1: Phylogenetic relationships: There are hundreds of proposed phylogenies on the evolution of actinopterygii, especially Teleostei. These references are ignored, and only the work of Parey et al. (2023) where Osteoglossomorpha + Elopomorpha are sister groups to Clupeocephala is cited. I remind the authors that only a dozen species were included in the analysis, so the results should be discussed with great caution. There is an enormous number of fossils and morphological synapomorphies pointing to other evolutionary scenarios.

Response 1: We are very grateful for the shortcomings you pointed out regarding the phylogenetic panel, and to supplement this section we have made the following corrections: we have added literature citations and discussions on different hypotheses of teleost phylogeny, and stated that our data support that Osteoglossomorpha + Elopomorpha are sister groups to Clupeocephala hypothesis. We removed the original overly absolute conclusions, indicating the tendency of the results in our dataset, which can be valuable for studying early teleosts phylogeny. Finally, we emphasize the value of phylogenetic analyses through early teleost fishes for understanding the evolution of teleosts origins.

Comments 2.: Biogeography: Based on the dating of the obtained topology, the authors claim that the evolution of teleosts has temporal congruence with the separation of continents. Well, since when did the continents stop moving? Continental drift is continuous; at no point in the geomorphological history of the planet has it been particularly greater. It is possible that the drift of continents affected the evolution of teleosts, but how did this happen? How can the authors confirm this relationship without testing any hypotheses? Thus, the results are discussed in overly speculative terms.

Response 2: Thank you for your suggestions. First, we recalculated the divergence times, and increased the number of iterations to ensure that the data at different divergence points converge. The new divergence time suggests that the teleost fishes originated at 269.04 Ma, while the divergence of the Osteoglossomorpha occurred at 202.78 Ma. (Line 191-192) Second, we have combined the analysis of geographic events with that of previous studies, which have shown the correlation between the movement of geographic plates and the divergence of fishes, e.g., the expansion of marine fishes is related to the movement of continental shelves. Our results favor this hypothesis to some extent. That is, plate drift may have influenced fish formation to some extent.

Comments 3: Evolution of limbs: I'm not an expert in Teleostei development, so I could contribute little to this last section. However, it is striking how superficially the authors address such a relevant topic. The author’s state: "we proposed that the absence of this CNE segment in teleost fishes may account for the loss of its primitive skeletal structure, and that conserved regulatory elements are important genetic innovations in the evolution of vertebrate limbs." But what primitive structures were lost? They don't explain anything.

Response 3: We thank you for pointing out the shortcomings, and in order to express the theme of the article more accurately, we have revised the whole manuscript as follows:

  1. We have reformulated the title of the article " The chromosome-scale genome of Chitala ornata illuminates the evolution of early teleosts ", which does not overemphasize the important biological question of the evolution of the pectoral fins of teleosts through the assembly of individual genomes. We emphasize that the assembly of early teleost fish genomes provides a powerful contribution to the understanding of the genomic features of teleosts.
  2. We have reorganized the narratives and ideas about the pectoral fin regulatory elements section, firstly, we have added the research and discussion about the evolution of teleosts pectoral fins, e.g., "line247-253", teleosts completely discarded the metapterygium. In contrast, the pectoral fins of early vertebrates are divided into three parts: propterygium, mesopterygium and metapterygium. Second, we have revised the description of our results to indicate that our comparative genomics data suggest that pectoral fin evolution in teleosts may be related to the loss of certain conserved regulatory elements of the genome, suggesting that this genomic phenomenon is worthy of further exploration.
  3. Finally, we believe that early teleost fishes are an important window into the genetic evolution of teleosts. The high-quality genome provides an important data resource for further studies on the origin of many physiological phenotypes in teleost fishes.

Comments 4: Finally, I emphasize that the publication of the species genome is quite important for understanding the osteoglossiforms’ evolution, but I am skeptical about the presented results as they lack theoretical basis for their propositions. I suggest to the authors that they reorganize the manuscript to highlight the contributions of the work within the reach of the obtained results, avoiding speculative conjectures.

Response 4: We appreciate your suggestions, and we have reorganized the headings, paragraph content, and for speculative conclusions in the new manuscript to present our topic more clearly. In summary, we hope that the assembly of a high-quality genome will provide a useful data resource for understanding early teleosts evolution.

Reviewer 2 Report

Comments and Suggestions for Authors

The authors assembled a chromosomal-level genome for the Clown featherback (Chitala ornata), and obtained the highest quality genome assembly for Osteoglossiformes to date,  and revealed a unique deletion of regulatory elements is adjacent to the Gli3 gene specifically in teleosts. So, it is deduced that the deletion might  be related with the specialized adaptation of their pectoral fins. It is a piece of interesting and nice work. The findings would provide valuable insights into evolution of fish fins development. However, in the manuscript, there are still some points needed to be verified before this manuscript become  acceptable as a publication in the Journal.

Major points:

1. The title is not appropriate since only the last one (3.4) of the results is related to the pectoral fins evolution.

2. The authors should provide more evidence or reasonable explanation for the conclusion that CNE near  Gli3 gene is really a regulatory of  pectoral fins development in fish, and how about the mechanisms whereby the CNE regulates pectoral fins development in fish ?

Author Response

Comments 1: The title is not appropriate since only the last one (3.4) of the results is related to the pectoral fins evolution.

Response 1: Thank you for your valuable feedback, which has guided us in refining the focus and clarity of our manuscript. We have updated the title to better reflect the core focus of our study and reorganized the whole manuscript. In the new version of the manuscript, we have updated the title to "The chromosome-scale genome of Chitala ornata illuminates the evolution of early teleosts ", We focus on exploring the genomic features of early teleost fishes through the construction of high-quality genomes, including genome size, the proportion of repetitive sequences, and the possible evolutionary implications of the loss of certain potential regulatory elements. We aim to provide a reliable source of high-quality genomic data for the exploration of teleosts.

Comments 2: The authors should provide more evidence or reasonable explanation for the conclusion that CNE near Gli3 gene is really a regulatory of pectoral fins development in fish, and how about the mechanisms whereby the CNE regulates pectoral fins development in fish?

Response 2: Thank you. We understand your concern about the content of the CNE section, and we have added a discussion of the CNE section "line236-240", and we have revised our conclusions to be overly absolute, and in the absence of specific experimental frameworks, based on the results of comparing genomics and DNase-seq data, we can only draw the conclusion that this CNE is a potential regulatory element, which suggests that genomic sequence changes have some relevance to pectoral fin evolution. Our theme is to provide high-quality genomic resources for studying early teleosts genome evolution and exploring the genetic basis of pectoral fin evolution.

Reviewer 3 Report

Comments and Suggestions for Authors

Teleosts, the most prolific vertebrates, have undergone significant physiological transformations of their pectoral fins, with early species like Osteoglossiformes offering insights into their adaptive evolution. This study presents a high-quality chromosomal-level genome assembly for the Clown featherback (Chitala ornata), revealing a unique deletion near the Gli3 gene linked to pectoral fin adaptation and highlighting the role of transposable elements in their evolutionary trajectory.

Major concerns-
Despite utilizing a substantial amount of PacBio HiFi CCS data, the assembly resulted in 126 contigs with a contig N50 of 32.78 Mb. While the anchoring of contig sequences to chromosomes improved the contiguity, with 94.78% of the assembly anchored to 21 chromosomes, the presence of 126 contigs still indicates incomplete assembly. This fragmentation could hinder downstream analyses and complicate interpretation.

While the study claims the C. ornata genome to be of the highest quality among published fish genomes of Osteoglossiformes, it lacks a comprehensive comparative analysis with other closely related species.

While the study identifies a conserved non-coding element (CNE) associated with pectoral fin evolution and suggests its regulatory role, the functional validation of this element is limited to computational analyses and DNase-seq data.

Author Response

Comments 1: Despite utilizing a substantial amount of PacBio HiFi CCS data, the assembly resulted in 126 contigs with a contig N50 of 32.78 Mb. While the anchoring of contig sequences to chromosomes improved the contiguity, with 94.78% of the assembly anchored to 21 chromosomes, the presence of 126 contigs still indicates incomplete assembly. This fragmentation could hinder downstream analyses and complicate interpretation.

Response 1: Thank you for pointing out the deficiencies in genome assembly transfer, we recognize that our genome is not yet at the level of telomere-to-telomere (T2T) assembly, and this will be our next phase in bringing a more complete genome to early teleost fishes. We would like to show that the high continuity genome of the newly assembled genome produced in this study is excellent in comparison to the early teleosts in terms of published genomes (Table S1 and Figure 1b). And it meets the requirements of the analysis in terms of continuity and completeness, and the small number of fragmented genomes will not affect the overall results of the analysis.

Comments 2: While the study claims the C. ornata genome to be of the highest quality among published fish genomes of Osteoglossiformes, it lacks a comprehensive comparative analysis with other closely related species.

Response 2: Thanks to your suggestion, we comprehensively selected the genomes of early ray-finned fishes, early teleost fishes, and Ostariophysan fishes for genome assembly quality comparisons. The results show that our genome continuity remains excellent and exceeds that of the vast majority of bony fishes (Table S1 and Figure 1b). We acknowledge that genome integrity assessments do not appear to be leading the way, but we emphasize that the use of assemblies with highly continuous genomes provides an important data resource for fish genome research.

Comments 3: While the study identifies a conserved non-coding element (CNE) associated with pectoral fin evolution and suggests its regulatory role, the functional validation of this element is limited to computational analyses and DNase-seq data.

Response 3: Thank you for your valuable suggestions. Regarding the concerns about the CNE section, we have made the following revisions: First, we have added a discussion of the CNE function and impact section "line233-240", indicating that the CNE is essential for vertebrate evolution. Second, we have revised the overly absolute conclusions in the manuscript. In the absence of specific experiments, we can only conclude that the CNE is a potential regulatory element based on the results of comparative genomics and DNase-seq data, which suggests that genome sequence alterations have some relevance to pectoral fin evolution. Finally, our theme is to provide a reliable resource of genomic data for the study of the early euryhaline genome and pectoral fin evolution.

Round 2

Reviewer 2 Report

Comments and Suggestions for Authors

The authors improved the manuscript much and it is now acceptable as a publication for the Journal, IJMS.

Reviewer 3 Report

Comments and Suggestions for Authors

the authors have made appropriate changes to the manuscript as suggested by the reviewer and it is ready to be accepted.